# Guiding Drug Repositioning for Cancers Based on Drug Similarity Networks

**DOI:** 10.3390/ijms24032244

**Published:** 2023-01-23

**Authors:** Shimei Qin, Wan Li, Hongzheng Yu, Manyi Xu, Chao Li, Lei Fu, Shibin Sun, Yuehan He, Junjie Lv, Weiming He, Lina Chen

**Affiliations:** 1College of Bioinformatics Science and Technology, Harbin Medical University, Harbin 150081, China; 2Institute of Opto-Electronics, Harbin Institute of Technology, Harbin 150001, China

**Keywords:** NSCLC, drug repositioning, drug similarity network, Random Walk with Restart

## Abstract

Drug repositioning aims to discover novel clinical benefits of existing drugs, is an effective way to develop drugs for complex diseases such as cancer and may facilitate the process of traditional drug development. Meanwhile, network-based computational biology approaches, which allow the integration of information from different aspects to understand the relationships between biomolecules, has been successfully applied to drug repurposing. In this work, we developed a new strategy for network-based drug repositioning against cancer. Combining the mechanism of action and clinical efficacy of the drugs, a cancer-related drug similarity network was constructed, and the correlation score of each drug with a specific cancer was quantified. The top 5% of scoring drugs were reviewed for stability and druggable potential to identify potential repositionable drugs. Of the 11 potentially repurposable drugs for non-small cell lung cancer (NSCLC), 10 were confirmed by clinical trial articles and databases. The targets of these drugs were significantly enriched in cancer-related pathways and significantly associated with the prognosis of NSCLC. In light of the successful application of our approach to colorectal cancer as well, it provides an effective clue and valuable perspective for drug repurposing in cancer.

## 1. Introduction

Cancer has become one of the most difficult diseases to overcome in the world today. However, the discovery as well as development of new and effective anti-cancer drugs is a time-consuming and expensive process with a high attrition rate [1,2]. Drug repositioning, the process of finding new indications for approved drugs, has emerged as a powerful alternative strategy for the development of novel anticancer drug candidates due to its economical, efficient and risk-free nature [3,4].

Systems biology enables the study of whole system components formed by the interactions between molecules, rather than just the characteristics of individual molecules [5,6]. This allows systems biology-based approaches to integrate and analyze different types of large-scale biological datasets [7,8]. A series of systems biology-based approaches to drug repositioning have been opened in the context of the ever-increasing volume of medical data [3,9]. In recent years, the understanding of the molecular basis of cancer development, progression and metastasis, as well as the mechanisms of treatment, has grown exponentially [10,11,12,13]. At the same time, the increasing number of cancer bioinformatics tools, such as network propagation algorithms, similarity metrics, and more, have been widely used in systems biology-based cancer drug repurposing studies [14,15,16]. Networks are a key feature in systems biology, providing the tractability and interpretability of interactions between hundreds or even thousands of molecules and network-based approaches to drug repositioning play an extraordinary role [17,18].

Network medicine represents a concept and paradigm in modern biomedical research, and its application to drug repositioning studies has become increasingly popular [19,20,21]. Networks in drug repurposing studies are mainly constructed in terms of drug or disease similarity as drug-disease association-centric and drug-centric [22,23,24]. The former uses known drug-disease relationships to construct relevant heterogeneous networks and employs methods, such as graph convolution, matrix factorization and matrix completion, to predict new drug-disease associations [25,26,27]. Completing a heterogeneous drug-disease network by drug and disease similarity metrics, Meng et al. proposed a neural collaborative filtering approach based on neighborhood interaction to predict new potential drugs for breast cancer and non-small cell lung cancer (NSCLC) [28]. Yang et al. successfully identified possible new indications for Levodopa, Doxorubicin, Amantadine and Flecainide by introducing bounded nuclear norm regularization methods into the network [29]. In the heterogeneous drug-disease network, Shagahyegh et al. developed a methodology based on an improved non-negative matrix factorization to excavate novel drug-disease interactions and had been successfully carried out in breast cancer [30]. When constructing a network centered on drug-disease interactions, there may be a significant loss of network-specific information when the drug and/or disease relationships are mixed indiscriminately in a heterogeneous network [22].

On the other hand, the drug-centric drug similarity networks rely on drug characteristics such as structure, side effects and targets of action, which in turn guide the repositioning of drugs [31,32]. Song et al. used spectral clustering to divide the integrated drug similarity network into clusters and annotated each cluster with anatomical therapeutic chemical (ATC) codes. Drug pairs with high similarity or drugs with unusual ATC codes in a given cluster would have the potential for repositioning [33]. By developing a network centrality-based drug prioritization method, Lucreţia selected azelaic acid as a possible antitumour drug in a weighted drug similarity network [34]. These drug network-based approaches have the potential to repurpose drugs to solve the high-investment-low-success dilemma of new drug development [35,36]. Methods related to drug-centric networks are usually constructed by considering only the characteristics of the drug itself to construct drug similarity networks, lacking the consideration of disease pathogenesis.

The vast complexity of cancer phenotypes and genotypes is synthesized into an outstanding conceptual framework in the form of the conceptualized “Hallmarks of Cancer” [37], and as more is known about the characteristics of cancer, it is desirable to gain more insight into drug repurposing to help convert old drugs into cancer treatment drugs [38]. The antihelminthic drug niclosamide induces anticancer effects by mediating Wnt, STAT3 and NF-κB, signaling pathways that are hallmarks of cancer invasion and metastasis [39,40,41]. Inflammation is a recognized hallmark of cancer for the progression of malignancy. Aspirin triggered proresolving mediators resolvins and lipoxins that resulted in inflammation inhibition and cancer regression stimulation [42,43].

The ATC classification system is a WHO multi-label categorization system that classifies drugs according to their therapeutic, pharmacological and chemical properties and is the most widely recognized drug classification scheme available. Based on the assumption that similar drugs will have similar medical indication areas, the ATC categorization system has been widely used in drug repurposing studies [44]. ATC code and drug were characterized by their similarity profiles and a drug-ATC code interaction network was constructed. Drug ATC code prediction was considered as a binary classification problem, applying a classifier to predict the unknown ATC codes of drugs [45]. Peng et al. predicted new ATC codes for drugs by constructing substructure & target-drug-ATC networks [46]. All the above demonstrated that ATC code reflects the therapeutic effect of drugs well and is an effective medium to evaluate the similarity of clinical treatment of drugs. The construction of the drug similarity network by combining the therapeutic properties of drugs characterized by ATC codes and the pathogenesis of diseases characterized by hallmarks of cancer could effectively guide the repurposing of drugs.

Here, we proposed an innovative strategy for the screening of repurposed drugs for cancer by considering both the characteristics of the drug itself and the association between the drug and the pathogenesis of the cancer. Based on the DrugBank database, a cancer-related drug similarity network was constructed by integrating the functional and clinical treatment similarities of the drugs. In the network, for a specific cancer, a drug-cancer correlation score was developed and calculated for each drug. For the top 5% of drugs, a further comprehensive analysis of stability and druggable potential was performed to screen potential repurposed drugs for the cancer. It would provide new ideas for drug repositioning studies for cancer and provide a boost to the treatment of cancers.

## 2. Results

### 2.1. Cancer-Related Drug Similarity Network

In this study, the similarity between drugs was assessed by both their functional and clinical therapeutic properties. First, 13,580 drugs were acquired from the DrugBank database, and 5625 drugs, and their targets were derived after filtering for targets that did not match the gene IDs in the hallmark gene set functional categories. The enrichment of these drugs in each of the 50 hallmark gene sets was analyzed by the hypergeometric test (see Methods section Functional similarity of the drugs), and a total of 3065 drugs were detected to be significantly enriched in the corresponding hallmark gene set functional categories. Within these, most of the drugs showed a smaller number of gene sets enriched (Figure 1A). Over 60% of the drugs were enriched in more than one hallmark gene set, and each drug was enriched in an average of two gene sets. The functional similarity between drugs was measured by whether they were enriched in the same gene set. This resulted in 629,214 similar drug pairs consisting of 3065 drugs. Among them, 83% of the drug pairs shared one hallmark gene set, and the number of hallmark gene sets that could be co-enriched between drugs was up to 8 (Figure 1B).

We found that 3150 of 13,580 drugs derived from the DrugBank database have ATC codes. Of these, 73% had only 1 ATC code, and 15% had 2 ATC codes (Figure 1C). The clinical therapeutic similarity between the 3150 drugs was measured based on the ATC codes at levels 2, 3 and 4 (see Methods section Clinical therapeutic similarity of the drugs). The 95th percentile of the empirical probability distribution function composed of all similarity values was used to estimate the significance threshold with a value of 0.67 (Figure 1D). Made up of 1923 drugs with targets 181,893 similar drug pairs were determined based on the clinical therapeutic similarity between the drugs.

Eventually, the cancer-related drug similarity network consisting of 4302 nodes and 807,391 edges was created, combining the functional and clinical therapeutic similarities of the drugs by taking the union of detected drug pairs (Figure 1E). The network’s degree ranged from 4 to 1612, with 5% of the nodes having a degree greater than 1000 and the average degree of the nodes in the network being 375 (Figure 1F). It was shown that the nodes in the cancer-related drug similarity network were closely connected. Of the 24 known therapeutic drugs (Appendix A) for NSCLC, 17 were similar to each other (Figure 1F).

### 2.2. Stable Drug Candidates in Unweighted Pattern

In the cancer-related drug similarity network, the drug-cancer correlation scoring method was used to recognize drug candidates. For NSCLC, under the unweighting pattern, the transition probability was set to 1 if the drugs were similar to each other and 0 otherwise. The 24 known therapeutic drugs from the DrugBank database were selected as a seed set to compute the drug-cancer correlation score of each drug in the network.

Here, the top 5% of drugs were regarded as having a similarity with the therapeutic drugs of NSCLC by running the drug-cancer correlation scoring algorithm. The leave-one-out cross validation was performed to calculate the recall rate of 24 known therapeutic drugs of NSCLC; 18 of these drugs were recalled in the corresponding identified top 5% of drugs with a recall rate of 0.75 (Figure 2A). Accordingly, the algorithm was workable with the unweighting pattern, and the top 5% of drugs were more likely to be associated with the treatment of NSCLC.

The top 5% of drugs identified varied in different seed set conditions. The delete-n-out strategy (see Methods Section Identification of Stable Drug Candidates) was implemented to screen for stable drug candidates against NSCLC by considering the drugs and their numbers that formed the seed set. Starting from n = 1, the number of seeds was removed in increasing order. When n = 12, the set of identified candidate drugs was no longer fully included in the result of removing one less seed.

The 192 drugs (top 5%) were screened when all known therapeutic drugs were used as seeds. For each specified number of seed deletions, multiple different sets of top drugs were generated. Of them, 193–217 and 204–294 drugs were captured as the top 5% by the delete-1-out and delete-12-out strategies, respectively (Figure 2B). Here, the frequency of each drug was counted, and the 95th percentile of the frequency distribution was treated as the threshold for drug candidate screening. For the delete-1-out and delete-12-out strategies, 24 and 9999 were the frequency thresholds, respectively (Figure 2C). As the deleted seeds gradually increased, the captured drug candidates progressively became fewer. The 174 and 43 drug candidates were captured corresponding to the delete-1-out and delete-12-out strategies. The final 43 drug candidates were defined as the stable drug candidates screened in the unweighting pattern. Each of the 43 stable drug candidates had a top ranking in all seed set conditions, with 98% of them ranking top 50 on average (Figure 2D). It demonstrated that, in the unweighting pattern, the identified stable drug candidates are not only stable but also have top rankings. These candidates have promise as drugs with novel therapeutic potential for NSCLC.

### 2.3. Stable Drug Candidates in Weighted Pattern

In the cancer-related drug similarity network constructed by integrating two similarities, the difference in the contribution of the two similarities may affect the screening of potentially repositionable drugs. Therefore, 54 weighted patterns reflecting different degrees of contribution were represented by applying the drug-cancer correlation scoring approach. For NSCLC, in each weighted pattern, the delete-n-out strategy was performed, and the intersection of the recognized drugs was selected. The overlap of all weighting patterns was identified as stable drug candidates.

The recalls of the leave-one-out cross validation for all weighted patterns were 0.75 (for the 64% weighting patterns) or 0.71 (for the 36% weighting patterns) (Figure 3A). Totally, the drug-cancer correlation scoring method for NSCLC had high recall rates in both weighting and unweighting patterns, which indicated that the strategy was reasonable and feasible.

When all known therapeutic drugs were taken as the seed set, 192–216 drugs (top 5%) were recognized through 54 weighting patterns (Figure 3B). The number of intersections for the top 5% drugs is 160, and its proportion in each weighted pattern is 0.74 to 0.83 (Figure 3C). Since the drugs detected in different weighted patterns are variable, the overlap of all patterns will be regarded as more possible candidates.

The delete-n-out strategy was implemented in each weighting pattern with a cutoff condition of deleting 12 seeds; 193–217 drugs were recognized as the top 5% by delete-1-out strategy in different weighting patterns, and 204 to 294 drugs were detected as the top 5% through delete-12-out (Figure 3B). For each weighting pattern, the frequency counts of the drugs in the multiple identified the top 5% of drugs by removing a specified number of seeds were performed, and the 95th percentile of the frequency distribution was chosen as the threshold for screening. The delete-1-out has a frequency threshold of 24 for all 54 weights, that is, for each weight, the top 5% of drugs that appeared consistently in all seed sets were used as candidates. The frequency thresholds ranged from 9195–10,000 for delete-12-out in different weighting patterns (Figure 3D); 162–201 and 32–101 drugs were identified by implementing the delete-1-out and delete-12-out strategies in 54 weighting patterns, respectively (Figure 3E,F). The drugs as the minimum set identified by performing the delete-12-out strategy in each weighting pattern were defined as candidates for that pattern. The final 20 candidates that served as the intersection of the candidates captured by all weighting patterns were considered as stable drug candidates screened in the weighting pattern (Figure 3G). Of the 20 stable drug candidates, 19 had an average ranking in the top 50 and 80% were in the top 30. In the weighting pattern, the identified drug candidates had the top ranking and stability as those in the unweighting pattern.

### 2.4. Potential Repositionable Drugs for NSCLC

By analyzing the stable drug candidates under the unweighted and multiple weighting patterns, 43 and 20 drugs were screened, respectively. The 19 shared drugs were treated as initial predicted drugs (Figure 4A).

The druggability of the initial predicted drugs was evaluated to recognize potential repositioned drugs for NSCLC. The gene effects of 94 NSCLC cell lines derived from the genome-wide CRISPR knockout screens are stored in the DepMap database. Genes with gene effect scores < −0.5 were accepted as essential for the survival of the cell line. Out of 53 targets of 19 initial predicted drugs, 11 genes were detected as druggable targets based on whether the gene was necessary for the survival of at least one cell line (Figure 4B). Genes *PPAT*, *HSPA8* and *EGFR* were essential for the survival of 62, 35 and 16 NSCLC cell lines, respectively (Figure 4C). Of these, EGFR was targeted by 4 drugs, ERBB2 was targeted by 3 drugs and 6 survival-essential genes were targeted by 1 drug (Figure 4D). Ultimately, 11 potential repurposable drugs for NSCLC targeting 11 druggable targets were predicted.

The ranking of 11 potential repositioned drugs in unweighting and weighting patterns was examined (Figure 4E). Overall, 11 potential repurposed drugs ranked top in all conditions. The average rankings were all less than 51, with 91% of drugs ranked in the top 25 on average and 73% of drugs in the top 20.

For NSCLC, among the 11 potential repositioned drugs, 10 drugs (90.91%) had evidence to verify their activity against NSCLC (Table 1). Panitumumab, a recombinant humanized anti-EGFR monoclonal antibody, was found to have significant tumor suppressive effects in NSCLC cell lines in an in vitro study [47]. A randomized phase III study concluded that panitumumab in combination with erlotinib plus bevacizumab was an efficient second-line treatment option for patients with NSCLC [48]. Neratinib binds to and irreversibly inhibits EGFR and human epidermal receptor 2 (HER2), which patients with specific EGFR mutant types of lung cancer may be sensitive to [49,50]. Neratinib also exerted anti-proliferative effects on HER2-altered NSCLC cell lines and showed potent tumor growth inhibitory activity in mouse xenograft models [51]. Although icotinib and olmutinib were not marketed therapeutic drugs for NSCLC, they were documented in the DrugBank database as being investigated for the treatment of NSCLC. The investigational drug icotinib, a novel EGFR—tyrosine kinase inhibitor, has shown encouraging efficacy in patients with advanced NSCLC who have failed previous chemotherapy. Olmutinib is a drug under investigation for the treatment of metastatic T790M mutation-positive NSCLC. Dasatinib inhibited migration and invasion and induced cell cycle arrest and partial apoptosis in NSCLC cell lines [52]. Phase I/II studies have shown that dasatinib in combination with erlotinib is safe and feasible for the treatment of NSCLC [53,54]. Trastuzumab emtansine is an anti-HER2 antibody-drug conjugate that showed a signal of activity in patients with HER2 overexpressing advanced NSCLC [55], and a phase II basket trial evaluated the activity of trastuzumab emtansine in patients with HER2 mutated NSCLC with a high response rate [56]. In vitro experiments have shown that pertuzumab, a humanized anti-HER2 monoclonal antibody, was effective against lung cancer cell growth by inhibiting HER2/HER3 signaling [57]. The triple combination of trastuzumab, pertuzumab and docetaxel was feasible and effective for HER2-mutated pretreated advanced NSCLC patients in a phase II study, with trastuzumab and pertuzumab being the repositionable drugs predicted in our study [58]. Binimetinib plus carboplatin and pemetrexed chemotherapy for non-squamous NSCLC in a phase I study with an investigator-assessed objective response rate of 50% [59]. An ongoing phase II PHAROS study would evaluate the antitumor activity/safety of binimetinib plus encorafenib in patients with BRAFV^600^-mutated NSCLC [60]. The combination of ruxolitinib and afatinib was observed modest clinical activity in patients with EGFR-mutated NSCLC from a phase Ib study [61]. Although there is no direct evidence that mitotane mediates the treatment of NSCLC, experiments have shown that its target, FDX1, could mediate the metabolism of lung adenocarcinoma and affected prognosis [62]. Its other targets, ESR1, AR and PGR, were also associated with the prognosis of NSCLC and AR was essential for the survival of NSCLC cell lines. In addition, the targets of mitotane were enriched in cancer progression-related pathways.

Over 60% of NSCLCs express EGFR, which has become an important therapeutic target for the treatment of these tumors [63]. 29% of all known therapeutic drugs target EGFR. Of the predicted potential repurposed drugs, 4 target EGFR and all are single-target drugs. This indicated that these drugs might have the potential to treat NSCLC. When the expression or mutation of a gene is found to be associated with patient prognosis, this is typically taken as evidence that the gene is an important driver of disease progression. Therefore, genes associated with prognosis are often considered as potential targets for therapeutic development [64,65]. So we studied the prognostic value of the corresponding 36 targets of other 7 drugs in NSCLC through two online databases, KM Plotter [66] and PrognoScan [67]. Log-rank *p*-values and univariate cox regression *p*-values were both thresholded below 0.05 in KM Plotter and adjusted to cox *p*-values < 0.05 in PrognoScan. All 7 drugs had targets (30 targets in total) shown to be significantly associated with NSCLC overall survival (OS) in at least 3 independent datasets (Figure 5). The results of the survival analysis suggested that the 7 drugs were promising for the treatment of NSCLC.

The deregulation of pathway activity underlies many human diseases, and drugs may aim to correct dysfunction through direct effects on the activity of specific proteins in the relevant pathways [68]. Known therapeutic targets were mainly enriched in 4 types of pathways: the NSCLC pathway, tumor progression-related pathways, cell process-related pathways and immune-related pathways (Figure 6) [69]. Our predicted potential repositioned drugs were also enriched in these categories of pathways. Among the 11 potential repurposed drugs, 9 were enriched in the NSCLC pathway (hsa05223) (Figure 6A) and 11 were enriched in the “Proteoglycans in cancer (hsa05205)”, “VEGF signaling pathway (hsa04370)”, etc. pathways involved in the development of cancer (Figure 6B). It is well known that the abnormal regulation of cell processes is closely related to the occurrence of cancer. The 10 potential repurposable drugs were enriched in pathways related to cellular processes, such as MAPK (hsa04010), PI3K-Akt (hsa04151), ErbB (hsa04012) and other signaling pathways (Figure 6C). Dysregulation of immune status in the tumor microenvironment plays an important role in cancer development and progression, and immunotherapy has emerged as a powerful clinical strategy for the treatment of cancer [70,71,72]. Among our predicted drugs, 7 were enriched in immune-related pathways, including the well-known PD-1 checkpoint pathway (hsa05235), where PD-1 inhibitors have become one of the indispensable treatments for NSCLC (Figure 6D). From a functional point of view, given that the potential repositionable drugs were enriched in the same class of pathways as the known therapeutic drugs of NSCLC, they may have the same therapeutic effect.

## 3. Discussion

Drug repurposing has been widely recognized as a promising tool for accelerating the drug discovery process, where network-based approaches are receiving increasing attention [17]. Existing drug–drug network-based approaches usually consider only the target, side effects and structure of the drug, without considering the clinical therapeutic effects exhibited by the drug. In this paper, we proposed a new drug repositioning strategy based on drug similarity network. Not only the clinical efficacy of the drug was considered, but also the innovative combination with the pathogenesis of cancer. The scoring algorithm developed enables global assessment of the correlation between drugs and specific cancers. The implementation of the stability screening strategy allows to reduce the dependence on the number of known therapeutic drugs for a cancer thus identifying drugs that are more likely to be associated with the cancer. Survival-essential genes may be promising candidates for anticancer drug targets [73]. Therefore, using survival-essential genes as a screening requirement may make the identified drugs more promising for repositioning to specific cancers. The druggable potential review using survival-essential genes identified 11 potentially repositionable drugs for NSCLC, 10 of which were well confirmed. Particularly, the survival-essential genes of NSCLC, EGFR, ERBB2 and SRC, are both targets of known therapeutic drugs and potentially repositionable drugs. The drugs identified by our strategy are expected to be repurposed in NSCLC.

To further explore the wide applicability of the drug repositioning strategy, we applied this approach to colorectal cancer (the whole set of findings was detailed in the Appendix A). Seven of the nine potentially repurposable drugs were supported by the corresponding literature (Appendix A). Survival analysis showed that the targets of the potential repositionable drugs were significantly associated with the prognosis of colorectal cancer (Appendix A). The targets of potential repositionable drugs and known therapeutics were mainly enriched in 3 categories of pathways, including colorectal cancer pathway (hsa05210) (Appendix A), tumor progression-related pathways (Appendix A) such as HIF-1 signaling pathway (hsa04066), Choline metabolism in cancer (hsa05231), and Rap1 signaling pathway (hsa04015), MAPK signaling pathway (hsa04010) and other cellular process-related pathways (Appendix A). It is worth noting that the network constructed from a pan-cancer perspective has the potential to not only identify potentially repositionable drugs against specific cancers, but also to capture drugs with broad antitumor activity. For example, icotinib and neratinib were identified and confirmed in both NSCLC and colorectal cancer.

Our strategy has some limitations. We utilized common hallmarks of cancer development and progression to characterize drug mechanisms of action, and in addition, multi-omics data such as genomics, proteomics and metabolomics of cancer could be used to integrate cancer specificity to characterize drug mechanisms of action. On the other hand, we considered the clinical therapeutic properties and mechanism of action of the drugs to construct a drug similarity network. The constructed network could be extended by considering the pathways of drug action, side effects, etc., which may make potentially repositionable drugs more practical for clinical applications.

In conclusion, the potential repurposable drugs identified with our drug repurposing strategy in a pan-cancer context were well confirmed. These drugs may offer new opportunities for therapeutic interventions in NSCLC or colorectal cancer. The proposed strategy has promising applications for drug repositioning in more types of cancer.

## 4. Materials and Methods

### 4.1. Data Sets

Information on the drugs and their targets, as well as the ATC codes of the drugs, were derived from the DrugBank database (https://go.drugbank.com/; version 5.1.8, released on 3 January 2021) [74]. Hallmark gene set functional categories for cancer were curated from the Molecular Signature Database (MSigdb; http://software.broadinstitute.org/gsea/msigdb; version 7.4, released on 2 April 2021) resource, consisting of 50 gene sets involving 4383 genes [75]. The Cancer Dependency Map portal (DepMap; https://depmap.org/portal/; version 21Q4, released on 3 November 2021) integrates CRISPR knockout screens published from Broad’s Achilles and Sanger’s SCORE projects, from which gene effect scores for cancer cell lines were derived. Negative scores represent that knockout genes inhibit cell growth, whereas positive scores represent those promote growth.

### 4.2. Methods

In order to explore and investigate in depth the issue of drug repurposing in cancer and to screen potential drug candidates for the treatment of a specific cancer, we implemented an integrated repurposable drug screening strategy (Figure 7). The cancer-related drug similarity network was constructed by assessing the functional and clinical therapeutic similarity of drugs based on the hallmark gene sets and the ATC codes of the drugs. For a specific cancer, a new algorithm was developed to evaluate the drug-cancer correlation score for each drug in the network, and the stability of the top-ranked drugs was further examined, which resulted in stable drug candidates. Finally, the stable drug candidates were analyzed for the druggable potential to identify repurposable drugs for cancer.

#### 4.2.1. Cancer-RELATED Drug Similarity Network Construction

For cancers, given that the pathogenesis is characterized by the instability of its hallmark genes, the therapeutic mechanism for cancer may be to correct the dysfunctional state of the disease by enriching the targets of its therapeutic drugs to hallmark gene set functional categories. Based on this assumption, we computed the functional similarity between drugs based on hallmark gene sets in which the drugs were enriched. In addition, the ATC system is an internationally accepted drug classification system based on therapeutics and chemistry, reflecting both the pharmacological and therapeutic effects of drugs. Therefore, the clinical therapeutic similarity between drugs could be estimated with the ATC codes. Finally, an integrated drug–drug similarity network related to cancer pathogenesis was constructed by combining similar drug pairs recognized based on the functional and clinical treatment similarities of the drugs.

##### Functional Similarity of the Drugs

Functional similarity between drugs was calculated based on the hallmark gene sets. A hypergeometric test analysis was performed on the drug $d_{x}$ to enrich its targets into gene set h of the 50 hallmark gene sets for cancers:(1)$$P_{h}\left(d_{x}\right)=1-\sum_{f=0}^{F-1}\frac{\left(\begin{array}{c} M\\ f \end{array}\right)\left(\begin{array}{c} Q-M\\ q-f \end{array}\right)}{\left(\begin{array}{c} Q\\ q \end{array}\right)}$$
where Q is the total number of genes in 50 hallmark gene sets (Q = 4383), q is the count of targets of drug $d_{x}$ contained in 50 hallmark gene sets, M is the number of genes in gene set h and F is the count of targets of $d_{x}$ included in h. A 50-dimensional vector $H\left(d_{x}\right)$ was used to represent the enrichment of drug $d_{x}$ in the 50 hallmark gene sets where $d_{x}$ was considered to be enriched in h if $P_{h}\left(d_{x}\right)\leq 0.05$ and was marked as 1 in the vector; otherwise, it was marked as 0. The functional similarity $S_{hallmark}\left(d_{i},d_{l}\right)$ of drug $d_{i}$ and drug $d_{l}$ was then estimated using the Jaccard index $J_{accard}\left(H\left(d_{i}\right),H\left(d_{l}\right)\right)$ of $H\left(d_{i}\right)$ and $H\left(d_{l}\right)$:(2)$$S_{hallmark}\left(d_{i},d_{l}\right)=\left\{ \begin{array}{l} 1,if\;J_{accard}\left(H\left(d_{i}\right),H\left(d_{l}\right)\right)\neq 0\\ 0,otherwise \end{array}\right.$$

If $S_{hallmark}\left(d_{i},d_{l}\right)$ is equal to 1, the two drugs are considered similar; otherwise, they are dissimilar.

##### Clinical Therapeutic Similarity of the Drugs

The ATC classification system is divided into five levels representing increasingly finer drug categories, in which levels 2, 3 and 4 codes stand for the therapeutic and pharmacological information of drugs [46]. Here, the clinical therapeutic similarity between drugs was estimated based on the 2nd-, 3rd- and 4th-level ATC codes. The $k_{th}$ level drug therapeutic similarity $\left(S_{k}\right)$ between drug $d_{i}$ and drug $d_{l}$ is defined as follows:(3)$$S_{k}\left(d_{i},d_{l}\right)=\frac{ATC_{k}\left(d_{i}\right)\cap ATC_{k}\left(d_{l}\right)}{ATC_{k}\left(d_{i}\right)\cup ATC_{k}\left(d_{l}\right)}$$
where $ATC_{k}\left(d_{x}\right)$ represents all ATC codes at the $k_{th}$ level of drug $d_{x}$. The score $S_{ATC}\left(d_{i},d_{l}\right)$ is used to define the therapeutic similarity between drug $d_{i}$ and drug $d_{l}$:(4)$$S_{ATC}\left(d_{i},d_{l}\right)=\frac{\sum_{k=2}^{k=4}S_{k}\left(d_{i},d_{l}\right)}{3}$$

The value of $S_{ATC}\left(d_{i},d_{l}\right)$ ranges from 0 to 1. The closer the value is to 1, the more clinical therapeutic similar the two drugs are. The significance threshold was then estimated based on the empirical probability distribution function (Pdf) of the similarity data among all drugs with ATC codes. If the similarity between drugs is not below this threshold, they are considered similar.

#### 4.2.2. Drug-Cancer Correlation Scoring

For a specific cancer, a new algorithm was designed for the global assessment of drug-cancer correlations in the cancer-related drug similarity network. The probability of each node was defined as the drug-cancer correlation score and used as a measure of how similar the node was to known therapeutic drugs. More specifically, Random Walk with Restart was performed in the network using the known therapeutic drugs for a cancer as the seed set, and after the seed set is propagated through, all nodes (the node representative drug in this study) in the network are ordered by the probability of arriving at the seed set by random walk. Accordingly, the drug-cancer correlation score for drugs in the network was defined as $S^{t+1}$:(5)$$S^{t+1}=\left(1-r\right)WS^{t}+rS^{0}$$
where restart probability r taken as 0.7, $S^{t}$ is a vector where its *i*-th element represents the probability of a random walker being at node i at time t, $S^{0}$ is the initial vector whose seed nodes were given equal probability with the sum of their probabilities of 1 while the other nodes were set to 0. W is the column normalized transition probability matrix consisting of transition probability $w_{ij}$ (from node i to node j). To reflect the importance of the two sources of similarity, the transition probability is measured by assigning weights. Thus, the $w_{ij}$ can be described as follows:(6)$$w_{ij}=\left\{ \begin{array}{l} 0\;or\;1,\;if\;unweighted\\ \alpha A+\beta B,if\;weighted \end{array}\right.$$

In the unweighting pattern, $w_{ij}$ is 1 if the two nodes are similar and 0 otherwise, and in the weighting pattern, A, B denote the functional and clinical treatment similarity of the drug, and α, β represent weighting coefficients for all combinations of integer values from 1 to 9 (the simplest ratio is taken if the α, β ratio is the same). A total of 54 combinations of weighting coefficients to be evaluated. The algorithm is considered to have converged when the difference between $S^{t+1}$ and $S^{t}$ (as measured by the L1 paradigm) is less than $10^{-10}$. The higher the drug-cancer correlation score of a drug, the higher the similarity to the seed set.

#### 4.2.3. Identification of Stable Drug Candidates

Both the drug-cancer correlation score and its ranking changed with the seed set. To identify stable drug candidates, we proposed a stability screening strategy by separately calculating the drug-cancer correlation scores for different seed sets formed by stepwise seed removal.

We implemented a strategy called delete-n-out (n is the number of seeds removed) by gradually eliminating seeds until the drug was unstable after frequency screening (Figure 7). More specifically, in the network, the number of deleted seeds grew from 1 to n in increasing order. With a given number of seeds removed, the drug-cancer correlation score was computed and ranked separately in descending order for each node in all different seed sets, and the top 5% of nodes were selected as potentially relevant for cancer treatment. The frequencies of the top 5% of nodes were then counted, and the 95th percentile of the frequency distribution was used as the lower bound to obtain the corresponding set of drug candidates. The candidate drug sets were recognized after comparing the removal of a different number of seeds. When the candidate drug set discriminated by removing one more seed that was completely included in the current candidate drug set, the process continued; otherwise, the deletion was stopped. In particular, when the number of seed combinations to be removed by the delete-n-out method was too large and very time-consuming due to the sheer number, we used the result of 10,000 randomizations as the candidate drug detected in that case. The ultimate set of identified drug candidates was used as the stable drug candidates. The pseudo-code for the entire stable drug candidate identification process is shown in Figure 8.

#### 4.2.4. Potential Repositionable Drug Screening

A two-step screening modality was proposed to identify potential repurposable drugs for a specific cancer. The first step was to evaluate the stable drug candidates identified by unweighted and different weighting patterns and used the intersection as initial predicted drugs $D_{p}$:(7)$$D_{p}=D_{unweighted}\cap D_{weighted}$$
where $D_{unweighted}$ represents the stable drug candidates derived without weighting, and $D_{weighted}$ is the candidates filtered with weighting, consisting of the intersection of the stable drug candidates screened in different weighting patterns:(8)$$D_{weighted}=\cap_{j=1}^{j=54}C_{m}\left(j\right)$$
where $C_{m}\left(j\right)$ is the stable drug candidates recognized at the *j*-th weighting pattern (see section Identification of stable drug candidates).

The next procedure was to assess the capability of the initial predicted drugs. The selection of potential targets for cancer can be achieved using the CRISPR knockout screens, and gene effects represent the impact of knocked out genes on the survival of cancer cell lines [76]. Typically, the cut-off value will be set to –0.5 for gene effect scores indicating the significant depletion of cell lines [77]. Thus, the possible druggable targets G for the cancer I is described as:(9)$$G\left(I\right)=\left\{g|s_{core}\left(g\right)\leq -0.5,s_{core}\left(g\right)\in S_{cores}\left(g\right)\right\}$$
where $S_{cores}\left(g\right)$ is the set of gene effect scores resulting from knocking out gene g in the cell lines of cancer I. Ultimately, potential repositionable drugs $D_{R}$ for the cancer I were the initial predicted drugs whose targets were included in the druggable targets G(I):(10)$$D_{R}=\left\{D|T\left(D\right)\cap G\left(I\right)\neq \emptyset \;and\;D\in D_{p}\right\}$$
where T(D) represents the target set of drug D.

## Figures and Tables

**Figure 1 ijms-24-02244-f001:**
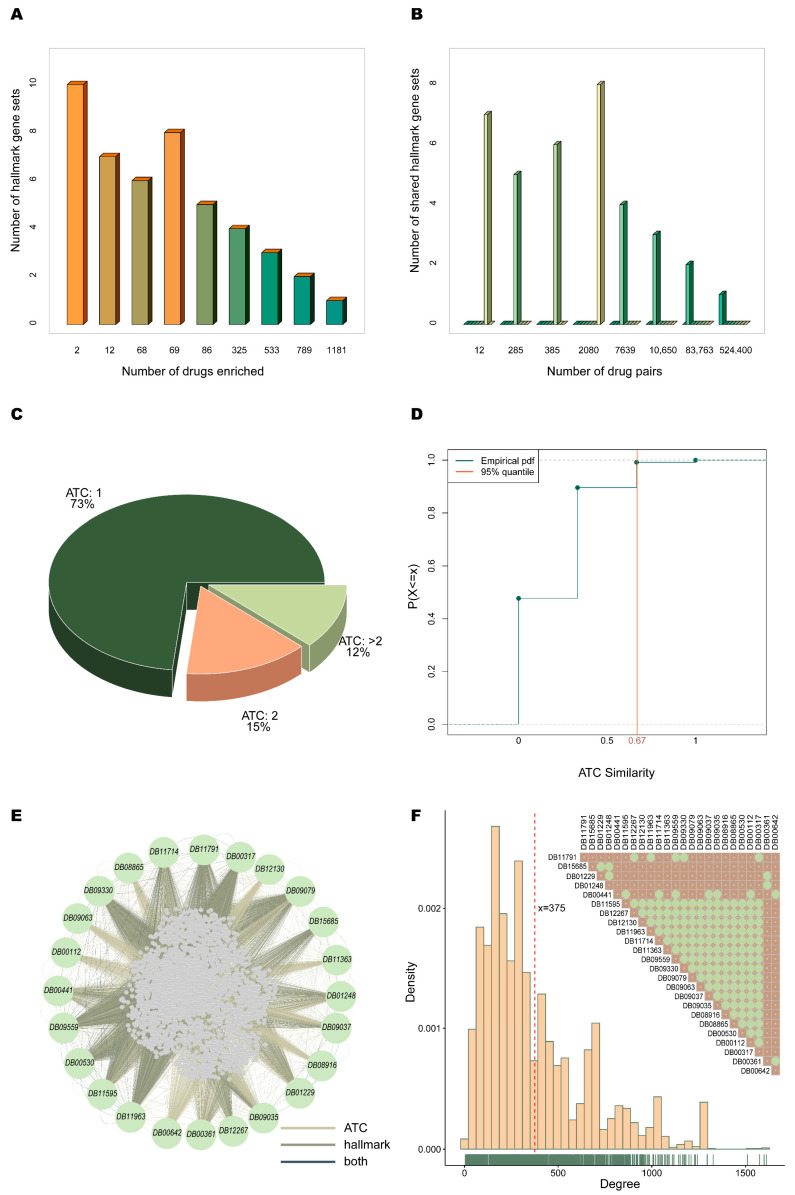
Cancer-related drug similarity network. (**A**) Number of hallmark gene sets that the drug is enriched to. (**B**) Number of hallmark gene sets shared by drug pairs. (**C**) Percentage of drugs with corresponding number of ATC codes. (**D**) Empirical probability distribution of clinical therapeutic similarity values. The orange vertical line represents the 95th percentile of similarity data with a value of 0.67. (**E**) Graph of the network (green nodes represent known treatments for NSCLC, the color of the edge represents the source of similarity). (**F**) Density plot of node degrees in the network. The carpet margin line represents the density of the distribution of data on the axes, and the red vertical line represents the mean of all node degrees with a value of 375. Additionally, the similarity plot between known therapeutic drugs of NSCLC is inserted, with bright green circles representing having similarity.

**Figure 2 ijms-24-02244-f002:**
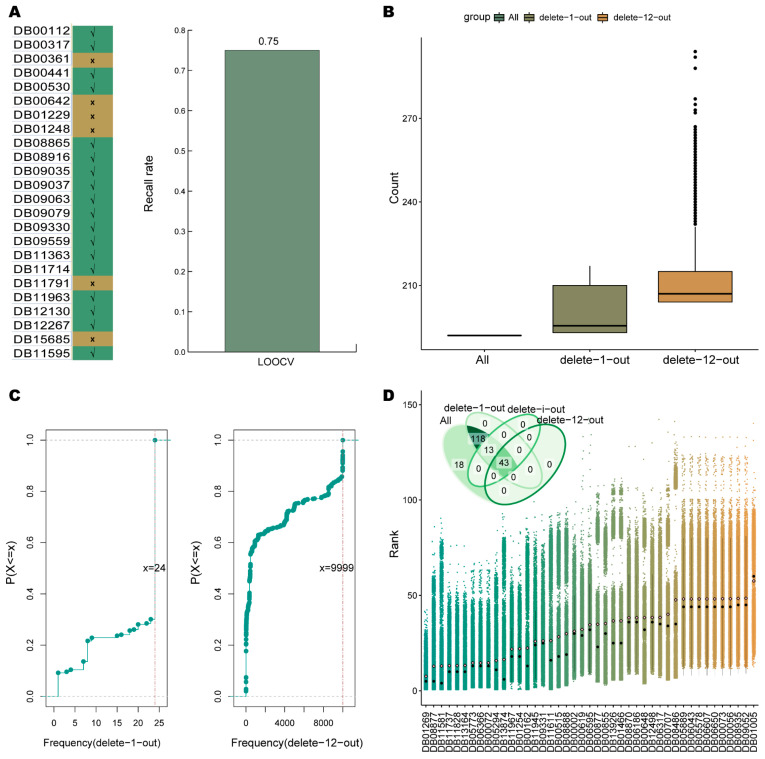
Stable drug candidates in unweighted pattern. (**A**) The recall of the leave-one-out cross validation. “√” means the drug was recalled in the top 5% of drugs, and “×” denotes that the drug was not recalled. (**B**) Distribution of the number of the top 5% of drugs. “ALL” represents all known therapeutic drugs as a seed set. (**C**) Frequency distribution. The orange dashed line shows the 95th percentile of the distribution. (**D**) Distribution of the ranking for the stable drug candidates in the unweighted pattern. Where the average ranking is marked by a pink diamond and the median is marked with a black dot. In addition, the Venn diagram of the top 5% of drugs filtered with different number of seeds removed is inserted.

**Figure 3 ijms-24-02244-f003:**
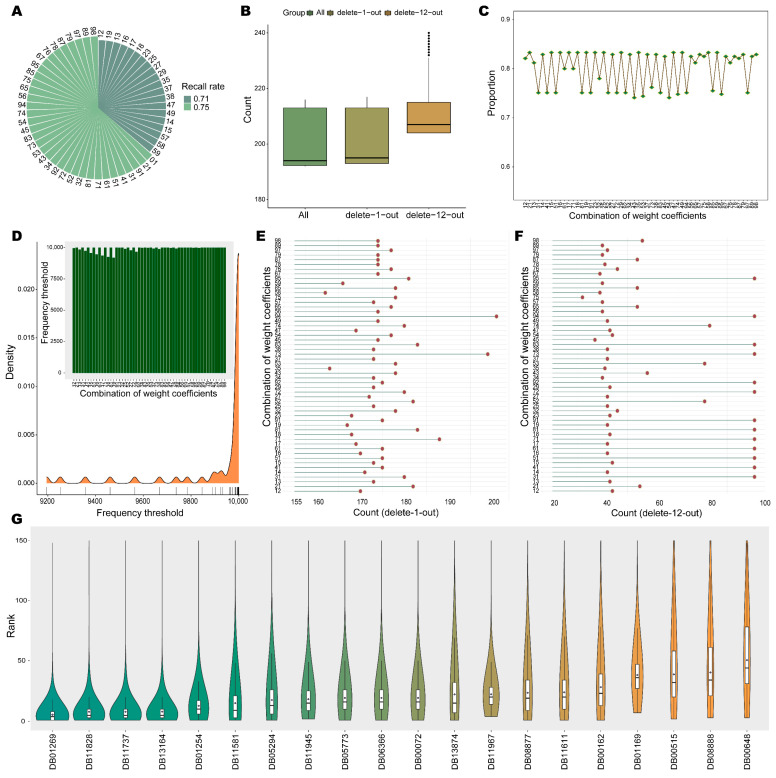
Stable drug candidates in weighted pattern. (**A**) The recall of the leave-one-out cross validation for each weighting pattern. (**B**) Distribution of the number of the top 5% of drugs. (**C**) The proportion of 160 intersecting drugs to the drug candidates identified by each weighted pattern. (**D**) Density curve of frequency thresholds under different weights by the delete-12-out strategy. Where the frequency threshold histogram corresponding to each weight is inserted. (**E**,**F**) The number of candidate drugs detected under different weights by the delete-n-out strategy. (**G**) Ranking distribution of stable drug candidates in all weighted patterns. Where the average rank is identified by a purple diamond.

**Figure 4 ijms-24-02244-f004:**
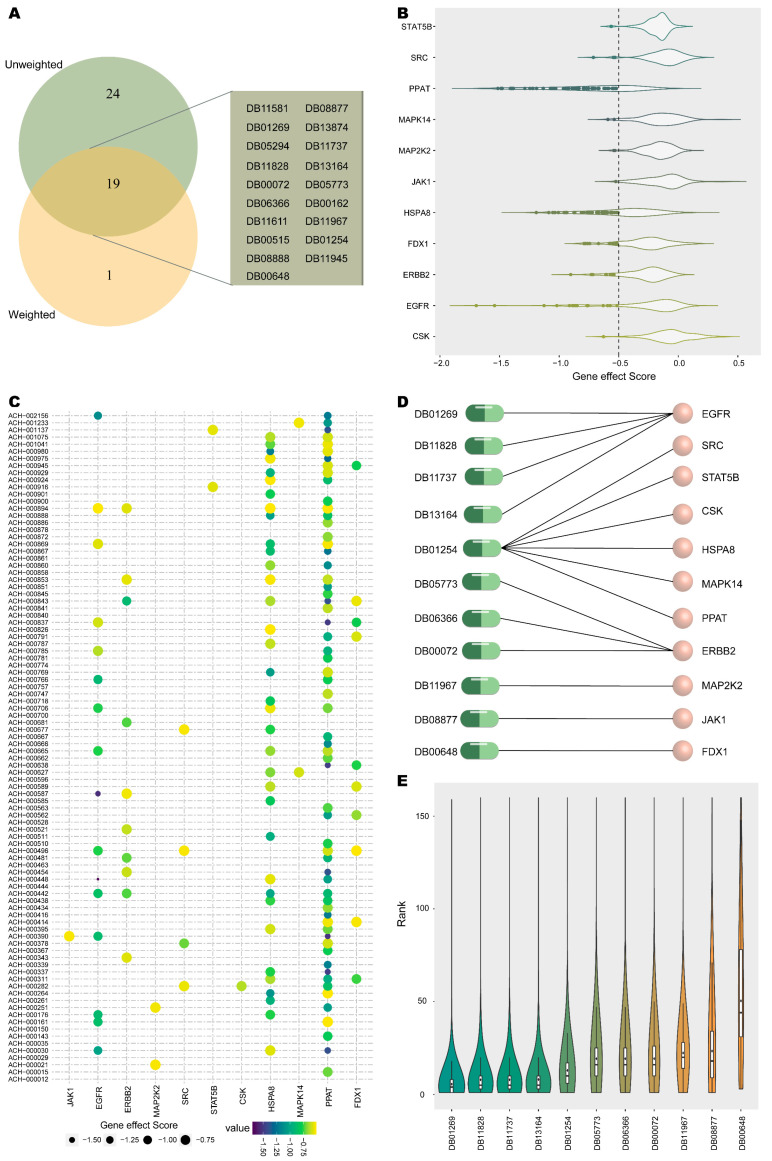
Potential repositionable drugs for NSCLC. (**A**) The overlap of stable drug candidates captured by weighted and unweighted patterns. (**B**) Distribution of survival-essential gene effect scores in NSCLC cell lines. Cell lines with gene effect <−0.5 are indicated by highlighted dots. (**C**) NSCLC cell lines corresponding to survival-essential genes. (**D**) Drugs targeting survival-essential genes. (**E**) Ranked distribution of potential repurposed drugs across all conditions, with the mean of the rank identified by the purple diamond.

**Figure 5 ijms-24-02244-f005:**
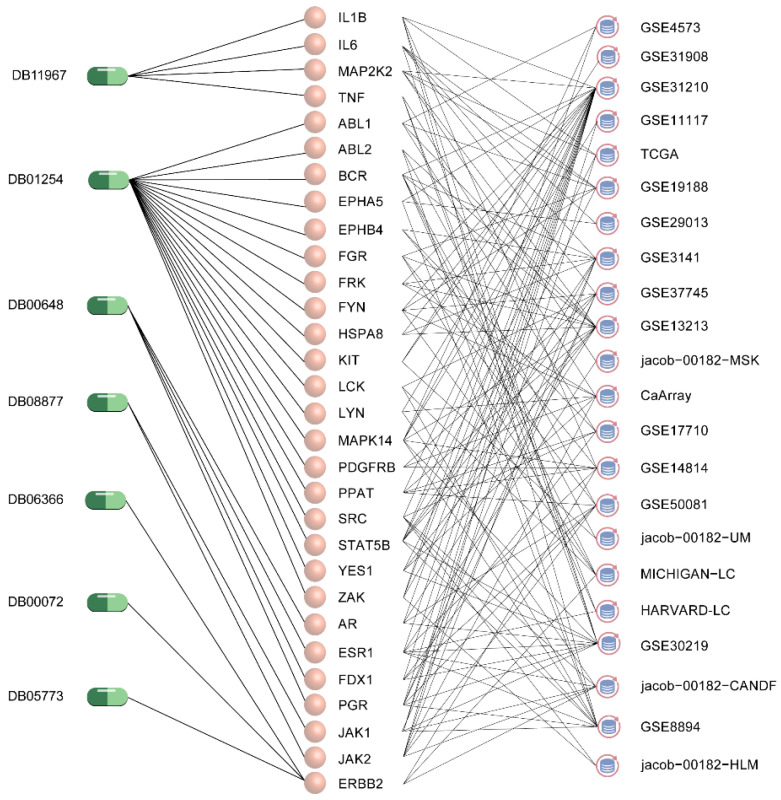
Survival analysis of potential repurposed drugs.

**Figure 6 ijms-24-02244-f006:**
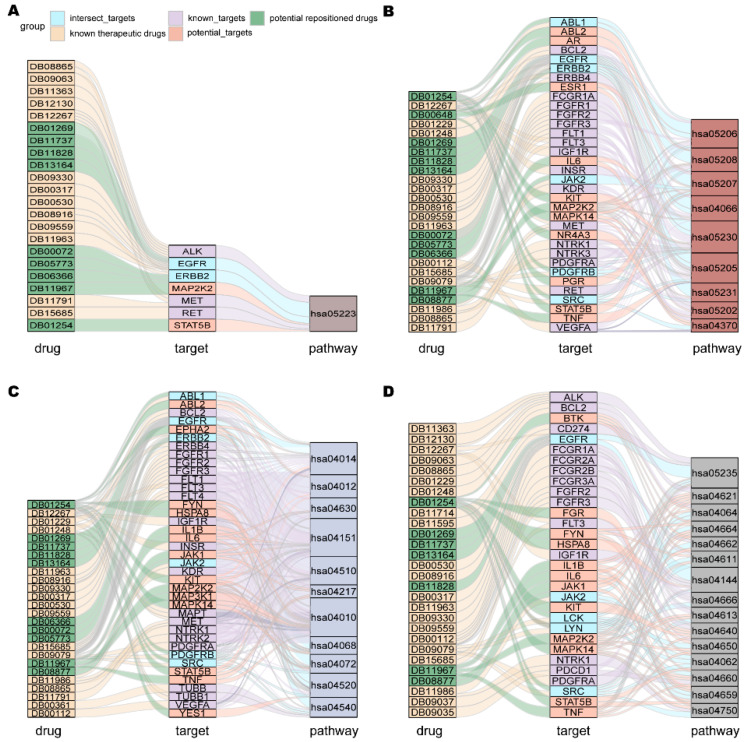
KEGG enrichment analysis. (**A**) NSCLC pathway. (**B**) Cancer progression-related pathway. (**C**) Cell process-related pathway. (**D**) Immune-related pathway. The different types of pathways are marked with different colors.

**Figure 7 ijms-24-02244-f007:**
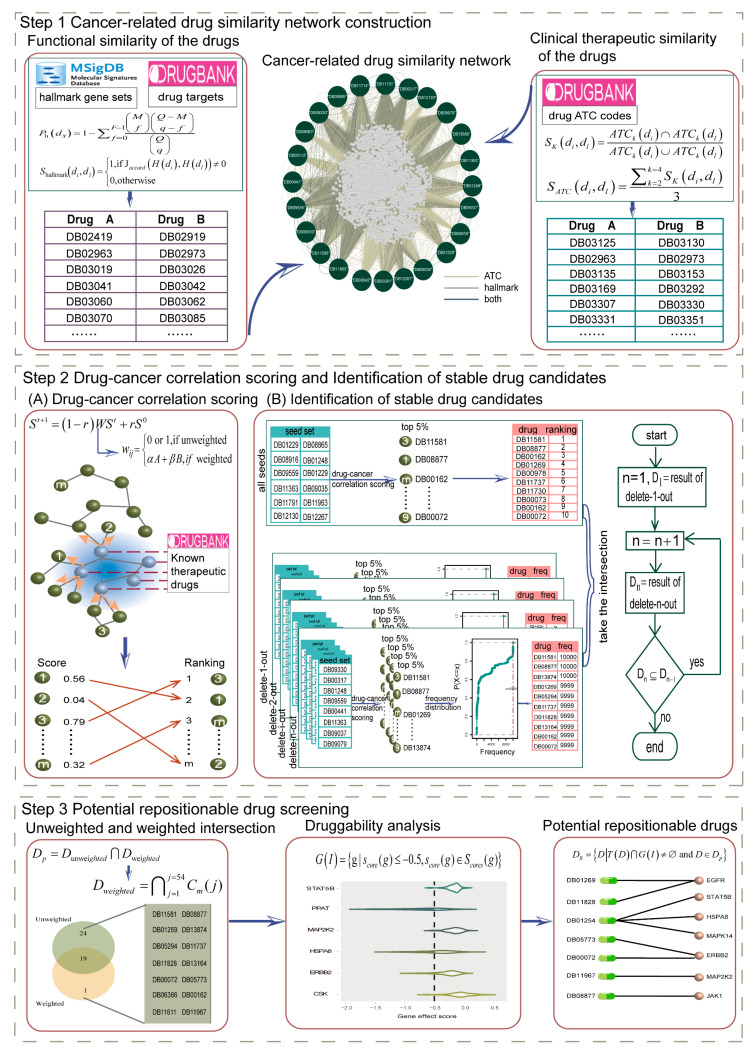
Flow chart of drug repurposing for specific cancer. Step 1, the functional similarity and clinical efficacy similarity between drugs are integrated to construct a cancer-related drug similarity network. Step 2, the correlation score of each drug with a specific cancer is calculated by using the known therapeutic drugs for the cancer as a seed set. The stability of the top 5% of drugs is examined by removing seeds in a stepwise increasing number. Step 3, the druggable potential of the shared stable drug candidates recognized in different weighting patterns is reviewed to identify potentially repositionable drugs for the cancer.

**Figure 8 ijms-24-02244-f008:**
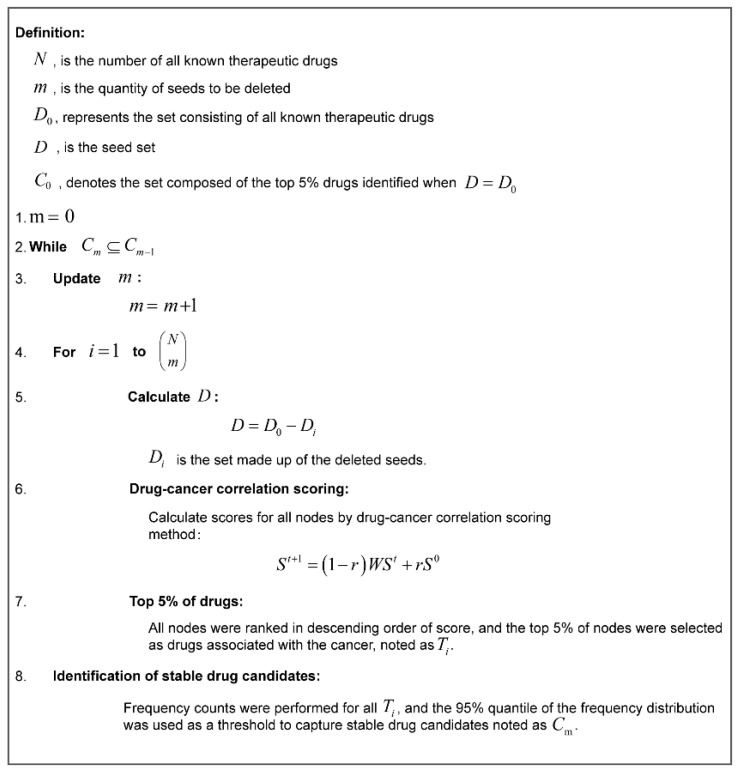
Pseudo-code for stable drug candidate identification.

**Table 1 ijms-24-02244-t001:** The potential repurposed drugs.

Average Rank	Accession Number	Drug Name	Evidence
6	DB01269	Panitumumab	[47,48]
8	DB11828	Neratinib	[49,50,51]
8	DB11737	Icotinib	DrugBank
9	DB13164	Olmutinib	DrugBank
14	DB01254	Dasatinib	[52,53,54]
21	DB05773	Trastuzumab emtansine	[55,56]
21	DB06366	Pertuzumab	[57,58]
21	DB00072	Trastuzumab	[58]
23	DB11967	Binimetinib	[59,60]
24	DB08877	Ruxolitinib	[61]
51	DB00648	Mitotane	Unconfirmed

## Data Availability

The data that support the findings of this study are available upon request.

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
