# Peer review of "Guiding Drug Repositioning for Cancers Based on Drug Similarity Networks"

_ijms, 2023, doi:10.3390/ijms24032244_

Round 1

Reviewer 1 Report

Reviewer Comments

You've come up with a novel approach to screening repurposed medications for cancer, one that takes into account both the drug's intrinsic properties and the link between the drug and cancer's pathogenesis. The DrugBank database was used to build a drug similarity network for cancer therapy, which took into account the medications' shared mechanisms of action and clinical efficacy. In the network, a drug-cancer correlation score was established and computed for each medication targeting that cancer. Those medications that made the top five percent of the list were subjected to a more thorough review of stability and druggable potential in order to test for possible repurposed pharmaceuticals to treat the malignancy. It would help cancer therapy by sparking fresh ideas for medication repositioning research.

1.     Please rewrite the abstract in a more readable way. In my view, this section may be strengthened by providing a clearer and more direct statement of the study's purpose.

2.     In the abstract, please describe the context and purpose of the investigation, then briefly discuss the methodology and findings, and conclude with your conclusion. However, take care not to confuse the reader.

3.     The section's introduction is significantly too short; the authors should include additional paragraphs and identify recent relevant papers. In addition, this section might utilize additional paragraphs to explain subjects such as the importance and utility of systems biology and cancer bioinformatics tools. I believe the reader will comprehend how to read the whole of your article.

4.     You may provide a brief description of the flowchart's activities in lieu of the highlighted statement "flow chart"

5.     Writers should generally check the English language and the overall structure of their work to avoid careless typos, missing commas, missing spaces between words, and cryptic sentences that should be changed to make them clearer.

6.     I believe it is preferable to include the whole set of findings in a supplementary file.

7.     Please present the databases and software you utilized in a short essay, and be sure to cite them.

8.     I believe you will be able to tackle the problems mentioned.

Reviewer 2 Report

Comment on an original paper

The original paper entitled “Guide drug repositioning for NSCLC based on drug similarity network” described the construction of the drug similarity network by combining the therapeutic properties of drugs and the pathogenesis of diseases to guide the repurposing of drugs. It is helpful for readers. The comment is as follows.

1. Line 70, antihelminthic drug. Do you mean clonidine or niclosamide because clonidine is an anti-hypertensive drug?

2. Lines 103-106, provide the released date of the data in MSigdb and DepMap.

3. Lines 118 or 194-198, it involved the stability screening. It will be more clear if you define the stable drug candidates.

4. Line 434, I suggest adding information on mitotane with unconfirmed evidence to NSCLS.

5. Line 511, give examples of the essential genes.

6. Lines 513-514 in the discussion, the author can give the drawback of the network, combining drug properties and cancer pathogenesis.

Reviewer 3 Report

Congratulations on the presented original work "Guide drug repositioning for NSCLC based on drug similarity network"; well written with the subject of matter of interest, Thanks for your work and your paper.
Analysing your paper, you must follow the Template and authors instructions, I would like to make a few suggestions:
- Regarding Figures, you should improve Figure 1 because it is difficult to read. You could present in Figure 1 only the highlighted genes and add as supplementary material the full figure. Figure 2 has small text legends hard to read and if this information was retrieved from a database, the reference should be presented and citations must be added.

 - Regarding Tables, you must improve because the texts are cut into pieces, so you should improve all tables.

Analysing the content, the paper is well written with a good structure and data presentation but in the discussion section must be improved.

Congratulations, again, thanks for your work and attention.
With my best regards.

Round 2

Reviewer 1 Report

Thanks the authors for the comprehensive answer.